# Quantum Policy Iteration via Amplitude Estimation and Grover Search – Towards Quantum Advantage for Reinforcement Learning

**Simon Wiedemann**\*                                                      *simonw.wiedemann@tum.de*
*Technical University of Munich, Germany*
**Daniel Hein**                                                             *hein.daniel@siemens.com*
*Siemens Technology, Munich, Germany*
**Steffen Udluft**                                                         *steffen.udluft@siemens.com*
*Siemens Technology, Munich, Germany*
**Christian B. Mendl**                                                     *christian.mendl@tum.de*
*Technical University of Munich, Germany*

**Reviewed on OpenReview:** *https://openreview.net/forum?id=HG11PAmwQ6*

## Abstract

We present a full implementation and simulation of a novel quantum reinforcement learning method. Our work is a detailed and formal proof of concept for how quantum algorithms can be used to solve reinforcement learning problems and shows that, given access to error-free, efficient quantum realizations of the agent and environment, quantum methods can yield provable improvements over classical Monte-Carlo based methods in terms of sample complexity. Our approach shows in detail how to combine amplitude estimation and Grover search into a policy evaluation and improvement scheme. We first develop quantum policy evaluation (QPE) which is quadratically more efficient compared to an analogous classical Monte Carlo estimation and is based on a quantum mechanical realization of a finite Markov decision process (MDP). Building on QPE, we derive a quantum policy iteration that repeatedly improves an initial policy using Grover search until the optimum is reached. Finally, we present an implementation of our algorithm for a two-armed bandit MDP which we then simulate.

## 1 Introduction

Successful reinforcement learning (RL) algorithms for challenging tasks come at the cost of computationally expensive training (Silver et al., 2016). Since quantum computers can perform certain complex tasks much faster than classical machines (Grover, 1996; Shor, 1999) a hope is that quantum computers might also enable more efficient RL algorithms. Today, there is theoretical and practical evidence that quantum interaction between agent and environment can indeed reduce the time it takes for the agent to learn (Section 2). In this work, we show that another possible source for quantum advantage in RL might come in the form of reducing the number of agent-environment interactions needed to find an optimal policy. We make this point by explicitly constructing a quantum policy iteration algorithm whose policy evaluation step is provably more sample efficient than a comparable classical Monte-Carlo (MC) approach.

In particular, we describe an optimization-based quantum algorithm for RL that makes use of a direct quantum mechanical realization of a finite MDP, in which agent and environment are modelled by *unitary operators* and exchange states, actions, and rewards in *superposition* (Section 4.1). A brief introduction to concepts from discrete quantum computing (including quantum superposition and unitary operators on

---

\*Work done while at Siemens.

quantum states) that are necessary to understand this construction is given in Section 3. Using the quantum realization of an MDP, we show in detail how to use amplitude estimation to estimate the value function of a policy in a finite MDP with finite horizon (Section 4.2). This quantum policy evaluation (QPE) algorithm can be compared to a MC method, i.e., sampling from agent-environment interactions to estimate the value function. QPE uses *quantum samples* (qsamples) instead of classical ones and provides a quantum advantage over any possible classical MC approach; it needs quadratically fewer qsamples to estimate the value function of a policy up to a given precision. We embed QPE into an optimization scheme that uses repeated Grover-like searches of the set of policies to find a decision rule with $\epsilon$-optimal performance (Section 4.3). The resulting routine shares similarities with the policy iteration method from *dynamic programming* (Sutton & Barto, 2018) and, thus, we call it *quantum (approximate) policy iteration.*

Our detailed description of quantum policy iteration shows how to solve a general MDP using only quantum methods and no classical sub-routines. We provide explicit constructions of all unitary operators used in our methods. The major contribution of our work is an in-depth description of QPE for finite MDPs and a proof of its quantum-advantage over classical MC. In the experiments, we exemplify QPE and quantum policy iteration on the level of single and multi qubit gates on a digital quantum computer. Simulations of this implementation numerically confirm the mathematically proven quantum advantage of QPE over classical MC policy evaluation and illustrate the convergence behavior of quantum policy iteration (Section 5).

## 2 Related Work

Most existing quantum RL approaches can be roughly divided into two categories: quantum enhanced agents that learn in classical environments, and scenarios where the agent and environment can interact quantum mechanically. For both approaches, there is evidence that quantum methods could indeed outperform classical approaches.

An example for the first category, is *quantum projective simulation* (Briegel & De las Cuevas, 2012), where an agent makes decisions using a quantum random walk on a learned graph that represents its internal memory. In this approach, the special properties of quantum walks allow for fast decision making and thus less active learning time than classical agents. Furthermore, various quantum-enhanced agents have been described that learn in classical environments by replacing components of classical learning algorithms with quantum counterparts. There are approaches for *deep Q-learning* and *policy gradient* methods, where classical artificial neural networks are replaced by analogous *variational quantum circuits* (VQCs) with trainable parameters (Chen et al., 2020; Lockwood & Si, 2020; Moll & Kunczik, 2021; Skolik et al., 2021; Jerbi et al., 2021).

Theoretical work by Dunjko et al. (2015) indicates that the possibility of quantum interaction between agent and environment can be leveraged to design quantum-enhanced agents that provably outperform the best possible classical learner in certain environments (Dunjko et al., 2015; 2016; 2017a). One such approach is related to our ideas and uses Grover searches in deterministic environments to find sequences of actions that lead to high rewards (Dunjko et al., 2016). The possibility of using amplitude estimation similar to QPE to be applied to stochastic environments are also discussed. Further extensions were proposed by Dunjko et al. (2017b) and Hamann et al. (2020).

Another related direction of quantum RL research is the development of methods based on DP. Naguleswaran & White (2005) propose to use quantum search methods to speed up a discrete maximization step during value iteration. Wang et al. (2021) used similar tools as in this work (i.e., amplitude estimation and quantum optimization) to obtain quantum speedups for subroutines of otherwise classical DP algorithms. Recently, Cherrat et al. (2022) developed a quantum policy iteration algorithm in which the Bellman equations for policy evaluation are solved using a quantum method for solving systems of linear equations.

Contrary to the methods by Naguleswaran & White (2005), Wang et al. (2021) and Cherrat et al. (2022), our novel quantum (approximate) policy iteration is not a quantum-enhanced version of an existing classical DP method and uses no classical subroutines. It also differs from the ideas of Dunjko et al. (2016) as it directly searches for the optimal policy instead of looking for rewarded actions that are then used to improve the decision rule.

## 3 Background

This work is largely concerned with quantum algorithms. These are routines that process information in the form of quantum states and follow the rules of quantum mechanics. To facilitate reading for readers with little or no background in quantum mechanics, we give a brief introduction to the most important concepts in this section. Thereby, we restrict ourselves to discrete quantum states which can be thought of as vectors in a finite-dimensional complex Hilbert space and operations on the states can be expressed using linear algebra. Discrete quantum mechanics is still a wide field, and we limit the content of this section to those concepts that are necessary to understand our work and omit most physical details. For a detailed introduction to quantum computing, we refer readers to the extensive textbook by Nielsen & Chuang (2002).

The quantum algorithms in this work process information on the states, actions and rewards of an MDP. If we want to find the best policy for an MDP on a classical computer, we would first have to encode the states, actions and rewards using classical bits and then use algorithms such as dynamic programming to determine the best policy. If we instead want to use a quantum computer, we first have to encode the states, actions and rewards as discrete quantum mechanical objects. Consider for example the set of all actions $\mathbb{A}$ of the MDP. We now discuss how to express this set and the elements therein in the language of quantum computing. Assume there are $N = |\mathbb{A}|$ actions. We identify each action $a_n \in \mathbb{A}$ with a member of an orthonormal basis of the Hilbert space $\mathbb{C}^N$, which we call the *action space* $\mathcal{A}$. For convenience, we choose the standard basis. This identification looks as follows

$$a_n \mapsto |a_n\rangle \,\hat{=}\, (0, ..., \underbrace{1}_{\text{index } n}, .., 0)^T. \tag{1}$$

The notation $|a_n\rangle$ emphasizes that this is a quantum state and considered a vector. Note that $\langle a_m | a_n \rangle = \delta_{m,n}$, where $\langle | \rangle$ denotes the inner product, and $\delta_{m,n}$ the Kronecker delta. Encoding MDP states and rewards works analogously and is described in Section 4.1.

So far, there is nothing quantum mechanical in our construction. This changes when we consider *quantum superpositions*. We again use the actions as an example. As we encoded each action as an orthonormal basis vector in a Hilbert space, we can consider linear combinations of actions such as for example

$$|\Psi\rangle = \sum_{n=1}^{N} c_n |a_n\rangle. \tag{2}$$

The state $|\Psi\rangle$ is a *superposition* of the basis actions $|a_1\rangle, ..., |a_n\rangle$. In order for $|\Psi\rangle$ to be a valid quantum state, the coefficients $c_n$ have to satisfy the normalization property

$$\sum_{n=1}^{N} |c_n|^2 = 1. \tag{3}$$

Note that this directly implies $\langle \Psi | \Psi \rangle = 1$, and every quantum state has to satisfy this condition. The reason for this requirement becomes evident when we discuss the concept of a *projective measurement*. The quantum state $|\Psi\rangle$ from Equation (2) is not classical information in the sense that even if it "exists" on a quantum computer, we cannot directly observe the coefficients $c_n$ that uniquely determine it. This is due to fundamental properties of quantum systems and we refer to more detailed introductions to quantum computing or mechanics (Nielsen & Chuang, 2002; Cohen-Tannoudji et al., 1977). The only way we can gain classical, human-interpretable information on the state $|\Psi\rangle$ is via *measurement*. This is a stochastic operation that returns any basis action state $|a_n\rangle$ with probability $|c_n|^2$. In this way, we can consider the state $|\Psi\rangle$ to be a quantum encoding of a probability distribution of the actions. In general, one can use the superposition principle to express any probability distribution over finitely many values as a quantum state. We will heavily use this fact in Section 4.1. As long as we do not measure the state $|\Psi\rangle$, it remains in superposition, which is what is often thought of as being in all of the states $|a_1\rangle, ..., |a_n\rangle$ "at the same time". The measurement process destroys the superposition and leads to a so-called *collapse*: If a measurement yields a basis action $|a_n\rangle$, the state of $|\Psi\rangle$ collapses to this measured basis action $|a_n\rangle$ and all other information that is encoded in the superposition is lost. We can think of measurement as sampling from the distribution encoded by

the coefficients $c_n$. Therefore, any superposition state can be considered a *quantum sample (qsample)* of the probability distribution it encodes. Via measurement, we can turn a qsample into a classical sample. Note that if we had access to M copies of the quantum state $|\Psi\rangle$, we could conduct M measurements and use the results to approximate the squared moduli of the coefficients $c_n$ via averaging.

A specific quantum system that is very important for quantum computing and which we also use in this work is the qubit. It is a two-dimensional system whose state space can therefore be thought of as $\mathbb{C}^2$. The basis states (basis vectors) are often denoted by $|0\rangle$ and $|1\rangle$ which reminds of the two classical states "0" and "1". A qubit can be regarded a generalization of a classical bit. In this sense, it is the smallest unit of information a quantum computer can process.

So far, we only discussed one quantum mechanical state in isolation. For our quantum mechanical realization of a classical MDP, we have to proceed to *joint quantum states* which describe multiple quantum mechanical systems simultaneously. While individual quantum states can be used to express and sample from probability distributions of a single random variable, joint states behave analogously for joint distributions. To give a concrete example for a joint state of two systems, consider the following scenario: Assume an agent starts an MDP at a random state where it chooses a random first action. This is determined by a set of probabilities $P(s, a)$ for all state-action pairs $(s, a) \in \mathtt{S} \times \mathtt{A}$. We now want to express this distribution in quantum mechanical terms. Following the description above, we encode the states and actions as basis vectors of a complex $\mathcal{S}$ and action space $\mathcal{A}$. The quantum analogue of a state-action tuple $(s, a)$ is a joint state

$$|s\rangle |a\rangle := |s\rangle \otimes |a\rangle \in \mathcal{S} \otimes \mathcal{A}, \tag{4}$$

where $\otimes$ denotes the Kronecker product. We say that this state has two *subsystems*, one for the state and one for the action. These two subsystems form the state-action space $\mathcal{S} \otimes \mathcal{A}$ which has, by properties of the Kronecker product, an orthonormal basis given by the set $\{|s\rangle |a\rangle\}_{s\in\mathtt{S},a\in\mathtt{A}}$ and is therefore $|\mathtt{S}| \cdot |\mathtt{A}|$ dimensional. We can express the distribution of initial states and actions as a superposition

$$|\Phi\rangle = \sum_{s\in\mathtt{S},a\in\mathtt{A}} c_{s,a} |s\rangle |a\rangle, \tag{5}$$

where the coefficients $c_{s,a}$ are any complex numbers that satisfy $|c_{s,a}|^2 = P(s, a)$. The joint state $|\Phi\rangle$, too, can be measured as described above. A measurement of the state $|\Phi\rangle$ in the $|s\rangle |a\rangle$ basis would yield a state-action pair with the corresponding probability. However, instead of measuring both subsystems, we can also measure only one of them. Say we measure the first subsystem. Then we observe a specific state $|s'\rangle$ with probability $\sum_{a\in\mathtt{A}} |c_{s',a}|^2 = \sum_{a\in\mathtt{A}} P(s', a)$. Therefore, measuring the first subsystem returns a sample of the *marginal* distribution $P(s)$ of the states that is induced by the joint distribution $P(s, a)$. Like above, this measurement also results in a collapse. Assume again that we observed a specific state $|s'\rangle$. This changes the sate of $|\Phi\rangle$ to

$$|\Phi'\rangle = \sum_{a\in\mathtt{A}} c_{a|s'} |s'\rangle |a\rangle, \tag{6}$$

where the coefficients of the superposition are given by

$$c_{a|s'} = \frac{c_{s,a'}}{\sqrt{\sum_{a\in\mathtt{A}} |c_{s',a}|^2}}. \tag{7}$$

Note that $|c_{a|s'}|^2 = P(a|s')$, which means that $|\Phi'\rangle$ is a quantum encoding of the conditional probability distribution $P(a|s')$. In summary, measuring the first subsystem does two things: It returns a (classical) sample of the states and leaves us with the conditional distribution of the actions given the observed state.

So far, we only discussed how the information we wish to process with the quantum computer is encoded. Now we focus on *how* the computer processes the information, i.e., how it changes quantum states. We already saw one possibility to modify a quantum state which is via measuring one or more subsystems. Apart from measurement, the only possibility a quantum computer has to change a quantum state is via *unitary quantum gates*. For discrete quantum mechanical systems, these gates correspond to unitary operators. The requirement of unitarity is important since it preserves the normalization of quantum states. Again, this is best understood by considering an example. For simplicity, consider a single qubit with its two-dimensional

state space. The *Hadamard* gate $\mathbf{H}$ (which we also use in Section 4.2) is defined as the linear extension of the mapping

$$|0\rangle \overset{\mathbf{H}}{\longmapsto} \frac{1}{\sqrt{2}}\big(|0\rangle + |1\rangle\big), \qquad |1\rangle \overset{\mathbf{H}}{\longmapsto} \frac{1}{\sqrt{2}}\big(|0\rangle - |1\rangle\big). \tag{8}$$

By identifying the basis states $|0\rangle$ and $|1\rangle$ with the canonical basis of $\mathbb{C}^2$, we can express the Hadamard gate $\mathbf{H}$ as a matrix, i.e.,

$$\mathbf{H} \triangleq \begin{pmatrix} 1 & 1 \\ 1 & -1 \end{pmatrix}. \tag{9}$$

It is easy to see that the Hadamard gate is a unitary operator. In general, quantum gates on n-dimensional systems correspond to unitary operators (matrices) that act on the n-dimensional state-space. In the following, we denote the group of unitary operators on a Hilbert space $\mathcal{H}$ by $U\big(\mathcal{H}\big)$. Analogously to quantum gates that act on individual systems, quantum gates on joint systems are unitaries on the Kronecker product of the quantum state-spaces of the systems. For example, a quantum gate that acts on the spaces $\mathcal{A}$ and $\mathcal{S}$ of MDP actions and states must be a member of $U\big(\mathcal{A} \otimes \mathcal{S}\big)$.

A special case of quantum gates on joint systems that play a central role in Section 4.1 and in many quantum algorithms in general are *controlled gates* which are also closely related to the concept of *entanglement*. We illustrate both concepts using the following example: Consider an MDP with two states and actions, i.e., $\mathbf{S} = \{s_0, s_1\}$ and $\mathbf{A} = \{a_0, a_1\}$. Assume the agent follows the deterministic policy $\pi$ that chooses action $a_0$ in state $s_0$ and action $a_1$ in state $s_1$, i.e., $\pi(a_0|s_0) = \pi(a_1|s_1) = 1$. We can describe this deterministic policy using a controlled quantum gate. More precisely, we will now discuss how to express the policy $\pi$ using a *controlled not (CNOT)* gate. This gate operates on the joint state of two qubits and its action is given by

$$|x\rangle \, |y\rangle \overset{\mathbf{CNOT}}{\longmapsto} \begin{cases} |x\rangle \, |y + 1 \bmod 2\rangle & \text{if } x = 1 \\ |x\rangle \, |y\rangle & \text{else} \end{cases} . \tag{10}$$

In other words: The CNOT gate flips the state of the second qubit whenever the first qubit, the so-called *control qubit* is in state $|1\rangle$ and leaves both states unchanged if the control qubit is in state $|0\rangle$. Now we return to our concrete example situation and the discrete policy. As the state and action spaces are two-dimensional, we can represent them using qubits, i.e.,

$$s_0 \mapsto |0\rangle_{\mathcal{S}}, \quad s_1 \mapsto |1\rangle_{\mathcal{S}}, \qquad \text{and} \qquad a_0 \mapsto |0\rangle_{\mathcal{A}}, \quad a_1 \mapsto |1\rangle_{\mathcal{A}}, \tag{11}$$

where the subscripts $\mathcal{S}$ and $\mathcal{A}$ emphasize that the two qubits come from the (separate) state and action Hilbert spaces. Using this identification, we see that the CNOT gate realizes precisely the action of the deterministic policy $\pi$. This is a concrete example of the *policy operator* which we define in Section 4.1.

Let us extend out example and consider the situation where the agent is in state $s_0$ and $s_1$ with equal probability and in each state chooses an action according to the deterministic policy $\pi$ that corresponds to the CNOT gate. As discussed above, the distribution of the agent's state can be encoded by the equal superposition state $\frac{1}{\sqrt{2}}\big(|0\rangle_{\mathcal{S}} + |1\rangle_{\mathcal{S}}\big)$. A direct calculation shows that

$$\frac{1}{\sqrt{2}}\big(|0\rangle_{\mathcal{S}} + |1\rangle_{\mathcal{S}}\big) \otimes |0\rangle_{\mathcal{A}} \overset{\mathbf{CNOT}}{\longmapsto} \frac{1}{\sqrt{2}}\big(|0\rangle_{\mathcal{S}} |0\rangle_{\mathcal{A}} + |1\rangle_{\mathcal{S}} |1\rangle_{\mathcal{A}}\big). \tag{12}$$

This means that if we apply the CNOT gate to a joint system that encodes the agent's random state, we get a qsample of the resulting probability distribution of state-action pairs that arise from the deterministic policy $\pi$. The state

$$|\Upsilon\rangle = \frac{1}{\sqrt{2}}\big(|0\rangle_{\mathcal{S}} |0\rangle_{\mathcal{A}} + |1\rangle_{\mathcal{S}} |1\rangle_{\mathcal{A}}\big), \tag{13}$$

is a so-called *Bell state* and is a famous example of the class of entangled states. Entanglement is a property of joint quantum systems and, loosely speaking, encodes dependency of quantum systems. In classical RL, the action of an agent is typically dependent on the state it is in. Note that here, too, the state $|\Upsilon\rangle$ encodes a dependent distribution of states and actions. In our quantum MDP construction in Section 4.1, we use controlled gates to produce entangled states that encode joint distributions of states, actions and rewards with stochastic dependencies. Entanglement is an important resource of many quantum algorithms such as the phase estimation algorithm (Figure 1) which we use for policy evaluation.

# 4 Quantum (Approximate) Policy Iteration

## 4.1 Quantum Mechanical Realization of a Finite Markov Decision Process

In this section, we use the concepts discussed in Section 3 to develop a quantum version of the classical MDP. The main difference between our construction and the classical one is that we use the superposition principle to model the stochasticity of the agent-environment interaction. Everything else is designed to be completely analogous to the classical case.

Consider a (classical) MDP with finitely many states $S$, actions $A$ and rewards $R$. As already described in Section 3, we can identify these sets with finite-dimensional complex Hilbert spaces $\mathcal{S}$, $\mathcal{A}$ and $\mathcal{R}$, which we call *state, action* and *reward spaces*. Formally, we use maps from the sets to the corresponding spaces, i.e.,

$$s \mapsto |s\rangle, \qquad a \mapsto |a\rangle, \qquad r \mapsto |r\rangle. \tag{14}$$

We require that the states, actions and rewards are orthonormal, i.e.

$$\langle s|s'\rangle = \delta_{s,s'}, \quad \langle a|a'\rangle = \delta_{a,a'}, \quad \langle r|r'\rangle = \delta_{r,r'}, \tag{15}$$

holds for all pairs $s, s' \in S$, $a, a' \in A$ and $r, r' \in R$. On a quantum computer that operates on qubits, we can for example arbitrarily enumerate all states and actions and use systems (strings) of qubits as quantum representatives, i.e., $s_i \mapsto |\text{bin}(i)\rangle$ and $a_j \mapsto |\text{bin}(j)\rangle$, where $\text{bin}(k)$ denotes the binary representation of an integer $k$. The rewards can be approximated by fixed point binaries which can also be represented as strings of qubits.

Agent and environment are modelled by unitary operators that act on $\mathcal{S}$, $\mathcal{A}$ and $\mathcal{R}$. For any policy $\pi$, we define the *policy operator*, as a unitary operator $\mathbf{\Pi} \in U(\mathcal{S} \otimes \mathcal{A})$ which satisfies

$$|s\rangle |0\rangle_{\mathcal{A}} \overset{\mathbf{\Pi}}{\longmapsto} |\pi_s\rangle = \sum_{a \in A} c_{a|s} |s\rangle |a\rangle, \tag{16}$$

for all states $s \in S$. The state $|0\rangle_{\mathcal{A}}$ is an arbitrary reference state in $\mathcal{A}$ and each amplitude $c_{a|s} \in \mathbb{C}$ has to satisfy $|c_{a|s}|^2 = \pi(a|s)$. In the language of Section 3, the policy operator can be described as a collection of controlled operators: For each MDP state, there is a controlled operator on $\mathcal{S} \otimes \mathcal{A}$ that prepares a qsample of the agent's policy in that state.

The policy operator can be chosen as any unitary operator that satisfies Equation (16). To see why such an operator always exists for any arbitrary stochastic policy $\pi$, note that Equation (16) describes a bijection between two orthonormal bases (ONBs) of two subspaces of $\mathcal{S} \otimes \mathcal{A}$. We can extend both ONBs to ONBs of the full space and define an arbitrary bijection between the newly added basis states while maintaining the action described in Equation (16) on the original ones. The linear extension of this assignment is then unitary by construction. Using an analogous argument, we define the *environment operator* $\mathbf{E} \in U(\mathcal{S} \otimes \mathcal{A} \otimes \mathcal{R} \otimes \mathcal{S})$ as any unitary operator that satisfies

$$|s\rangle |a\rangle |0\rangle_{\mathcal{R}} |0\rangle_{\mathcal{S}} \overset{\mathbf{E}}{\longmapsto} \sum_{r,s'} c_{r,s'|s,a} |s\rangle |a\rangle |r\rangle |s'\rangle, \tag{17}$$

with amplitudes that satisfy $|c_{r,s'|s,a}|^2 = p(r, s'|s, a)$ for all state-action pairs $s \in S$, $a \in A$. A single interaction between agent and environment is modelled by the *step operator* $\mathbf{S} := \mathbf{E} \circ \mathbf{\Pi}$. For any state $s \in S$, it holds that

$$|s\rangle |0\rangle_{\mathcal{A}} |0\rangle_{\mathcal{R}} |0\rangle_{\mathcal{S}} \overset{\mathbf{S}}{\longmapsto} \sum_{a,r,s'} c_{a,r,s'|s} |s\rangle |a\rangle |r\rangle |s'\rangle, \tag{18}$$

where $c_{a,r,s'|s} = c_{r,s'|s,a} c_{a|s}$. We use $\mathbf{S}$ to construct a unitary operator that prepares a quantum state which represents the distribution of all trajectories with a fixed finite horizon $H$. We call it *MDP operator* and denote it as $\mathbf{M}$. While the step operator acts on $\mathcal{S} \otimes \mathcal{A} \otimes \mathcal{R} \otimes \mathcal{S}$, the MDP operator is a unitary on the *trajectory space* $\mathcal{T}^H := \mathcal{S} \otimes (\mathcal{A} \otimes \mathcal{R} \otimes \mathcal{S})^{\otimes H}$ which is large enough to store $H$ quantum agent-environment interactions. The states in this space represent trajectories of length $H$ and are of type

$$|t^H\rangle = |s_0^{t^H}\rangle |a_0^{t^H}\rangle |r_1^{t^H}\rangle |s_1^{t^H}\rangle \cdots |r_H^{t^H}\rangle |s_H^{t^H}\rangle. \tag{19}$$

We define the MDP operator as

$$\mathbf{M} := \prod_{h=1}^{H} \mathbf{S}_h \in U(\mathcal{T}^H), \tag{20}$$

where $\mathbf{S}_h$ denotes a local version of $\mathbf{S}$ that acts on the $h$-th $\mathcal{S} \otimes \mathcal{A} \otimes \mathcal{R} \otimes \mathcal{S}$ subsystem of $\mathcal{T}^H$. It holds that

$$|s\rangle \otimes \left( |0\rangle_{\mathcal{A}} |0\rangle_{\mathcal{R}} |0\rangle_{\mathcal{S}} \right)^{\otimes H} \overset{\mathbf{M}}{\longmapsto} \sum_{t^H} c_{t^H} |t^H\rangle, \tag{21}$$

where $|c_{t^H}|^2 = p(t^H)$ which is the classical probability of trajectory $t^H$. This can be seen via inductive application of the relation given in Equation (18). Note that by construction, the state from Equation (21) is a qsample of the trajectories of length $H$.

For QPE, we need qsamples of the returns in order to approximate the value function. With this in mind, we use the qsample of the trajectories and calculate for each trajectory state $|t^H\rangle$ the associated return defined as $G(t^H) := \sum_{h=1}^{H} \gamma^{h-1} r_h$, where $\gamma \in [0,1]$ is a fixed discount factor. As the reward states of $|t^H\rangle$ are qubit binary encodings of (real) numbers, we can use quantum arithmetic to calculate their discounted sum. Formally, we use a unitary *return operator* $\mathbf{G}$ that maps

$$|r_1\rangle \cdots |r_H\rangle |0\rangle_{\mathcal{G}} \overset{\mathbf{G}}{\longmapsto} |r_1\rangle \cdots |r_H\rangle \left| \sum_{h=1}^{H} \gamma^{h-1} r_h \right\rangle, \tag{22}$$

where $|0\rangle_{\mathcal{G}}$ is a reference state in the *return space* $\mathcal{G}$, which represents a system of sufficiently many qubits to encode all returns. Ruiz-Perez & Garcia-Escartin (2017) show an explicit construction of such an operator which performs the weighted addition in the Fourier domain. By letting $\mathbf{G}$ act on all reward subsystems and the return component of $\mathcal{T}^H \otimes \mathcal{G}$, we can define $\mathbf{G} \circ \mathbf{M}$ which satisfies

$$|0\rangle \overset{\mathbf{G} \circ \mathbf{M}}{\longmapsto} |\psi\rangle := \sum_{t^H} c_{t^H} |t^H\rangle |G(t^H)\rangle. \tag{23}$$

As all $|t^H\rangle |G(t^H)\rangle$ states are (by construction) orthonormal, it follows that if we measure the second subsystem of $|\psi\rangle$, we receive an individual return with the classical probability determined by the MDP and the agent's policy. Therefore, (the second subsystem of) the state $|\psi\rangle$ is as qsample of the return.

## 4.2 Quantum (Approximate) Policy Evaluation

Consider the problem of (approximately) evaluating the value function $v_\pi^H$ of a policy $\pi$ for some finite horizon $H \in \mathbb{N}$. The value function is defined pointwise via

$$v_\pi^H(s) = \mathbb{E}_{t^H}[G(t^H)|S_0 = s], \tag{24}$$

where $s \in \mathtt{S}$ is the initial state, and the expectation is taken with respect to the distribution of all trajectories of length $H$. Even if we assume perfect knowledge of the MDP dynamics $p$, evaluating $v_\pi^H(s)$ exactly is usually infeasible: the complexity of directly calculating the expectation grows exponentially in the horizon $H$. Techniques from DP such as *iterative policy evaluation* reduce the computational complexity but come at the cost of high memory consumption (Sutton & Barto, 2018). An alternative approach is to use an MC method to estimate $v_\pi^H(s)$. The arguably most straightforward MC algorithm for policy evaluation is to collect a dataset of trajectories and to average the returns. This requires the possibility to sample from the distribution of the trajectories.

Montanaro (2015) developed a quantum algorithm to estimate the expectation of a random variable using qsamples. The method is quadratically more sample efficient than the best possible classical MC routine and it forms the foundation of QPE. To approximate the expected return, we use qsamples of the returns to which we have access via the operator $\mathbf{A}_{\mathrm{QPE}} := \mathbf{G} \circ \mathbf{M}$ from Equation (23). The first step towards QPE is to note that we can encode $v_\pi^H(s)$ as amplitude of a basis vector in a quantum superposition. To this end,

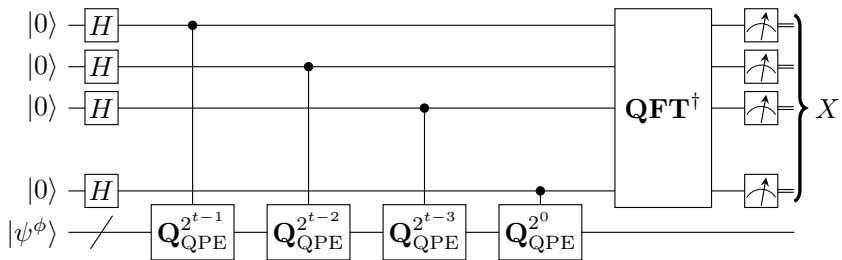

Figure 1: Phase estimation circuit for amplitude estimation in QPE.

assume that we know a lower bound $\underline{g}$ and an upper bound $\bar{g} \neq \underline{g}$ on all returns. Using an affine function $\phi$ defined by $\phi(x) = (x - \underline{g})/)(\bar{g} - \underline{g})$, we construct a unitary operator $\mathbf{\Phi}$ that maps

$$|x\rangle |0\rangle \xrightarrow{\mathbf{\Phi}} |x\rangle \left( \sqrt{1 - \phi(x)} |0\rangle + \sqrt{\phi(x)} |1\rangle \right), \tag{25}$$

where $|x\rangle$ is a binary (qubit) representation of a real number $x \in \mathbb{R}$ and the second subsystem is a single qubit. If we now apply $\mathbf{\Phi}$ to the return component of $|\psi\rangle$ from Equation (23) and an additional qubit, we get

$$|\psi\rangle \xrightarrow{\mathbf{\Phi}} |\psi^\phi\rangle := c_0 |\psi_0^\phi\rangle + c_1 |\psi_1^\phi\rangle, \tag{26}$$

where

$$|\psi_1^\phi\rangle \propto \sum_{t^H} c_{t^H} \sqrt{\phi\big(G(t^H)\big)} |t^H\rangle |G(t^H)\rangle |1\rangle, \tag{27}$$

The state $|\psi_0^\phi\rangle$ contains all the states in which the last qubit is in state $|0\rangle$ and is therefore orthonormal to $|\psi_1^\phi\rangle$. The amplitude $c_1 \in \mathbb{C}$, which is the norm of the (non-normalized) state on the right-hand side of Equation (27), satisfies

$$|c_1|^2 = \sum_{t^H} p(t^H)\phi\big(G(t^H)\big) = \mathbb{E}_{t^H} \left[ \phi\big(G(t^H)\big) \right]. \tag{28}$$

By affinity of $\phi$, it holds that

$$\phi^{-1}\big(|c_1|^2\big) = \mathbb{E}_{t^H} \left[ G(t^H) \right] = v_\pi^H(s). \tag{29}$$

Therefore, approximately evaluating the value function can be done by estimating $|c_1|^2$. This is a task which is commonly referred to as *amplitude estimation*.

One way to estimate $|c_1|^2$ (or any amplitude in general) is to encode it via the phases of two eigenvalues of a unitary operator and to estimate them using *phase estimation*. Following the original construction of Brassard et al. (2002), we define this unitary as

$$\mathbf{Q}_{\mathrm{QPE}} := -\mathbf{A}_{\mathrm{QPE}}^\phi \circ \mathbf{S}_0 \circ \big(\mathbf{A}_{\mathrm{QPE}}^\phi\big)^\dagger \circ \big(\mathbf{id}_{\mathcal{T}^H \otimes \mathcal{G}} \circ \mathbf{Z}\big), \tag{30}$$

where $\mathbf{A}_{\mathrm{QPE}}^\phi := \big(\mathbf{id}_{\mathcal{T}^H} \otimes \mathbf{\Phi}\big) \circ \mathbf{A}_{\mathrm{QPE}}$. In this expression, $\mathbf{\Phi}$ acts on the return subsystem and the ancillary qubit. The operator $\mathbf{S}_0$ is a *phase oracle* that, considering the basis states, flips the phase of a state precisely when all components (in this case the trajectory and the return component as well as the additional qubit) are in the corresponding $|0\rangle$ ground states. The $\mathbf{Z}$ operator corresponds to a Pauli-Z gate and therefore the last operator is also a phase oracle which flips the phase of a state precisely when the ancillary qubit is in state $|1\rangle$. Brassard et al. (2002) showed that $|\psi^\phi\rangle$ can be decomposed as $|\psi^\phi\rangle = c_+ |\psi_+^\phi\rangle + c_- |\psi_-^\phi\rangle$ where $|\psi_+^\phi\rangle$ and $|\psi_-^\phi\rangle$ are orthonormal, $|c_+|^2 = |c_-|^2 = 1/2$ and $\mathbf{Q}_{\mathrm{QPE}} |\psi_\pm^\phi\rangle = e^{\pm 2\phi i\theta} |\psi_\pm^\phi\rangle$, which means that $|\psi_\pm^\phi\rangle$ are eigenstates of $\mathbf{Q}_{\mathrm{QPE}}$. The phase $\theta$ is the unique value in $[0, 1)$ that satisfies $\sin^2(\pi\theta) = \sin^2(-\pi\theta) = |c_1|^2$.

Therefore, we can use a *phase estimation* algorithm to estimate $\theta$ and $-\theta$. The classical phase estimation routine is described in Figure 1. The figure shows a *quantum circuit* which is a representation of a quantum algorithm. Starting from the left, the quantum states are processed with the unitary gates illustrated as

boxes. Here, $\mathbf{H}$ denotes the Hadamard gate discussed in Section 3 and all systems except of $|\psi^\phi\rangle$ are qubits. The black dots at the qubits and the connection between them and the $\mathbf{Q}_{\mathrm{QPE}}$ gates denote that the $\mathbf{Q}_{\mathrm{QPE}}$ gates are controlled by the corresponding qubits (c.f. Section 3). The last boxes behind the $\mathbf{QFT}^\dagger$ (*quantum Fourier transform*) gate stand for quantum measurements. For a detailed analysis of the phase estimation algorithm, we refer readers to the textbook by Nielsen & Chuang (2002).

We see that the phase estimation circuit from Figure 1 uses a system of

$$t := n + \left\lceil \log_2 \left( \frac{1}{2\delta} + \frac{1}{2} \right) \right\rceil, \tag{31}$$

qubits where $n \in \mathbb{N}$ and $\delta \in (0, 1]$ are arbitrary. The basis states of these qubits are interpreted as binary numbers. Phase estimation exploits the fact that $|\psi^\phi_\pm\rangle$ are eigenstates of $\mathbf{Q}_{\mathrm{QPE}}$ to bring the $t$ qubits into a joint state $X$ which, when measured, satisfies

$$P\Big( |X/2^t - \theta| \leq 1/2^{n+1} \Big) \geq 1/2 - \delta/2, \tag{32}$$

$$P\Big( |X/2^t + \theta| \leq 1/2^{n+1} \Big) \geq 1/2 - \delta/2, \tag{33}$$

i.e., the distribution of $X/2^t$ is concentrated at $\theta$ and $-\theta$. In case $t = n$, we get the same result with total probability at least $8/\pi^2$. These bounds follow from a detailed analysis of the phase estimation circuit as was done for example by Cleve et al. (1998).

As $|c_1|^2 = \sin^2(\pi\theta)$ and $v_\pi^H(s) = \phi^{-1}(|c_1|^2)$, we define the QPE approximation $\tilde{v}_\pi^H(s)$ of $v_\pi^H(s)$ for any realization $x$ of $X$ as

$$\tilde{v}_\pi^H(s) := \phi^{-1}\Big( \sin^2\big(\pi x/2^t\big) \Big). \tag{34}$$

The approximation error of $\theta$ given in Equation (32) can be translated to an approximation error of the value function

$$P\big(|\tilde{v}_\pi^H(s) - v_\pi^H(s)| \leq \epsilon\big) \geq 1 - \delta, \tag{35}$$

where the error $\epsilon$ is (as shown in the Appendix) given by

$$\epsilon = (\bar{g} - \underline{g})\big(\pi/2^{n+1} + \pi^2/2^{2n+2}\big) \in \mathcal{O}(1/2^n). \tag{36}$$

Phase estimation uses $\mathcal{O}(2^t) = \mathcal{O}(1/\epsilon \cdot 1/\delta)$ applications of $\mathbf{A}^\phi_{\mathrm{QPE}}$ and $\big(\mathbf{A}^\phi_{\mathrm{QPE}}\big)^\dagger$ (via $\mathbf{Q}_{\mathrm{QPE}}$) to achieve this error bound (c.f. Figure 1). Each such application corresponds to the collection (preparation) of one qsample. According to the optimality results on MC methods due to Dagum et al. (2000), the sample efficiency of the best possible classical MC type algorithm to estimate $v_\pi^H(s)$ via averaging has sample complexity in $\Omega(1/\epsilon^2 \cdot \log_2(1/\delta))$. Therefore, QPE yields a quadratic reduction of the sample complexity with respect to the approximation error $\epsilon$. This quantum advantage holds over the best possible classical MC approach for policy evaluation. An advantage over other approaches is not guaranteed. Comparing the sample complexity of QPE to those of other classical methods is an interesting direction for future research. For more discussion on the exact nature of the quantum advantage of QPE and the conditions under which it holds, we refer the reader to the conclusion in Section 6.

### 4.3 Quantum Policy Improvement

The task of improving a given policy $\mu$ is to find another policy $\mu'$ such that $v_{\mu'}^H(s) \geq v_\mu^H(s)$ holds for all $s \in \mathtt{S}$ and that strict inequality holds for at least one $s' \in \mathtt{S}$. Once we know the value function $v_\mu^H$ of $\mu$, such a $\mu'$ can be explicitly constructed via *greedification*. In this section, we derive *quantum policy improvement* (QPI) which instead uses a Grover search over a set of policies to find an improved decision rule. For simplicity, we assume from now on that the MDP always starts in some fixed initial state $s_0 \in \mathtt{S}$. In this case, instead of using the whole value function, we can restrict ourselves to the *value* of policy $\pi$ which, under abuse of notation, we define as $v_\pi^H := v_\pi^H(s_0)$.

QPI uses Grover search on QPE results to find an improved decision rule. For a fixed policy $\pi$, QPE can be summarized as one unitary $\mathbf{QPE}$ (consisting of the phase estimation circuit from Figure 1 without the

measurements) that maps $|0\rangle \xmapsto{\textbf{QPE}} |\psi_{\text{QPE}}^{\pi}\rangle$. Now consider the finite set P that contains all deterministic policies. By mapping each $\pi \in$ P onto a member $|\pi\rangle$ of an ONB of some suitable Hilbert space $\mathcal{P}$, we can construct a unitary, $\pi$-controlled version $\textbf{QPE}_\pi$ of $\textbf{QPE}$ that satisfies

$$|\pi'\rangle |0\rangle \xmapsto{\textbf{QPE}_\pi} \begin{cases} |\pi\rangle |\psi_{\text{QPE}}^{\pi}\rangle & \text{if } \pi' = \pi \\ |\pi'\rangle |0\rangle & \text{else} \end{cases}, \tag{37}$$

for all policies $\pi' \in$ P. Using these operators, we define another unitary that prepares the search state for QPI Grover search. Consider

$$\textbf{A}_{\text{QPI}} := \left( \prod_{\pi \in \text{P}} \textbf{QPE}_\pi \right) \circ (\textbf{H}_\mathcal{P} \otimes \textbf{id}), \tag{38}$$

where $\textbf{H}_\mathcal{P}$ is a Hadamard transform on $\mathcal{P}$. By construction, this operator maps

$$|0\rangle_\mathcal{P} |0\rangle \xmapsto{\textbf{A}_{\text{QPE}}} \frac{1}{\sqrt{|\text{P}|}} \sum_{\pi \in \text{P}} |\pi\rangle |\psi_{\text{QPE}}^{\pi}\rangle. \tag{39}$$

As the $|\psi_{\text{QPE}}^{\pi}\rangle$ states encode the value of policy $\pi$, we can define a suitable oracle and use it to amplify the amplitudes of those policies that are likely to yield higher returns. For a policy $\mu$ and an approximation $\tilde{v}_\mu$ of its value, consider a phase oracle $\textbf{O}_{>\tilde{v}_\mu}$ that satisfies

$$|x\rangle \xmapsto{\textbf{O}_{>\tilde{v}_\mu}} \begin{cases} -|x\rangle & \text{if } \phi^{-1}\big( \sin^2(\pi x / 2^t) \big) > \tilde{v}_\mu \\ |x\rangle & \text{else} \end{cases}. \tag{40}$$

Using this oracle, we define the QPI Grover operator as

$$\textbf{Q}_{\text{QPI}}^{\tilde{v}_\mu} := -\textbf{A}_{\text{QPI}} \circ \textbf{S}_0 \circ \textbf{A}_{\text{QPI}}^{\dagger} \circ (\textbf{id}_\mathcal{P} \otimes \textbf{O}_{>\tilde{v}_\mu}). \tag{41}$$

According to the principle of *amplitude amplification*, which is used in Grover search and was described in detail by Brassard et al. (2002), if we apply this operator a certain number of times to the search state prepared by $\textbf{A}_{\text{QPI}}$, this amplifies the amplitude of all $|\pi\rangle |x\rangle$ states whose $|x\rangle$ component satisfies the oracle condition $\phi^{-1}\big( \sin^2(\pi x / 2^t) \big) > \tilde{v}_\mu$. As a result, the probability of measuring a policy that has better performance than $\tilde{v}_\mu$ (up to an error of $\epsilon$) is increased. Due to the QPE failure probability of $\delta$, the procedure may also amplify amplitudes of policies that do not yield an improvement (within $\epsilon$) as QPE may overestimate their performance. However, the probability of such errors can be limited by choosing small values for $\delta$.

For the amplitude amplification to work, we need to specify the number of *Grover rotations*, i.e., the number of times we apply $\textbf{Q}_{\text{QPE}}^{\tilde{v}_\pi}$, which determines how the amplitudes of the desired states are scaled. Although there is a theoretically optimal number of Grover rotations, this value is inaccessible in QPI as it would require knowledge of the probability of obtaining a desired state when measuring the search state (Brassard et al., 2002). To circumvent this problem, we use the *exponential Grover search* strategy introduced by Boyer et al. (1998): We initialize a parameter $m = 1$. In each iteration, we uniformly sample the Grover rotations from $\{0, ..., \lceil m-1 \rceil\}$ and measure the resulting state. If the result corresponds to an improvement, we reset $m \leftarrow 1$. Otherwise, we overwrite $m \leftarrow \lambda m$ for some $\lambda > 1$ which stays the same over all iterations. For a theoretical justification of this technique, please refer to the original work of Boyer et al. (1998).

Quantum (approximate) policy iteration as described in Algorithm 1 starts with an initial policy $\pi_0$ and repeatedly applies QPI to generate a sequence of policies with increasing values. Note that the procedure is similar to a quantum minimization algorithm introduced by Dürr & Høyer (1996). In each iteration $k$, quantum policy iteration runs QPI which is a Grover search and can therefore also fail to produce a policy whose estimated value exceeds the current best $v_k$. In this case, the policy and its value are not updated. As we typically do not know the value of the optimal policy, the algorithm uses a patience criterion to interrupt the iteration if there was no improvement in the last $C$ steps.

As quantum policy iteration relies on QPE, it can only find a policy with $\epsilon$-optimal behavior and may also fail to do so. This is because QPE yields an $\epsilon$-approximation of the value only with probability $1 - \delta$. We

---

**Algorithm 1** Quantum Policy Iteration

---

**input:** A policy $\pi_0 \in P$, an estimate $\tilde{v}_{\pi_0}$ of its value, parameters for QPE and QPI
**output:** Guesses $\tilde{\pi}^* \in P$ and $\tilde{v}^*$ for $\pi^*$ and $v^*$
$c \leftarrow 0$
**for** k=1,2,3,... **while** $c \leq C$ **do**
    Run QPI on $\pi_{k-1}$, $\tilde{v}_{\pi_{k-1}}$ to obtain $\tilde{\pi}_k$, $\tilde{v}_k$
    **if** $\tilde{v}_{\pi_k} > \tilde{v}_{\pi_{k-1}}$ **then**
        $\pi_k \leftarrow \tilde{\pi}_k$, $v_k \leftarrow \tilde{v}_k$, $c \leftarrow 0$
    **else**
        $\pi_k \leftarrow \pi_{k-1}$, $v_k \leftarrow v_{k-1}$, $c \leftarrow c+1$
    **end if**
**end for**
**return** $\tilde{\pi}^* = \pi_T$, $\tilde{v}^* = v_T$ from last iteration $T$

---

conjecture that the failure probability of quantum policy iteration can be made arbitrarily small by choosing a sufficiently small $\delta$ which, however, comes at the cost of increasing the qubit complexity and run-time of QPE. Moreover, note that if in some iteration $k$ the QPE output $v_k$ overestimates $v_{\pi_k}$ but still $v_k < v_*$, QPI may nevertheless find a policy $\pi_{k+1}$ with $v_{\pi_{k+1}} > v_k$. This mitigates QPE errors.

We propose that the complexity of quantum policy iteration is best measured in terms of the total number of Grover rotations performed in all QPI steps. This number is proportional to the run-time of the procedure and also measures the number of times we ran QPE which relates via the MDP model to the total number of quantum agent-environment interactions. Therefore, the number of Grover rotations also measures the qsample complexity. We conjecture that the complexity grows linear in $\sqrt{|P|}$. Our intuition behind this is that quantum policy iteration is essentially Grover search except that it uses changing, inaccurate oracles.

## 5   Experiments

Existing quantum hardware is not yet ready to run QPE, let alone quantum policy iteration. Both methods use far more gates than what is possible with existing, non-error-corrected digital quantum computers. Therefore, we have to resort to simulations. However, as quantum circuits are known to be hard to simulate classically, the complexity of the RL problems we can consider is limited. Simulating QPE alone requires processing a state vector whose size grows exponentially with the horizon, which is infeasible for large environments that require many interactions. For example: For a 10x10 maze problem with four actions and horizon $H$, there are $400^H$ trajectories which have to be encoded in a single state vector in order to accurately simulate QPE. Processing such enormous arrays is infeasible even for short horizons. Considering these limitations, we resorted to a two-armed bandit problem. This setup is far too simple to use it to fully test our methods. Rather than that, the idea behind the simulation study in this section is to illustrate the steps needed to realize our algorithms on real quantum hardware in detail and to demonstrate their behavior.

A two-armed bandit can be thought of as a slot machine with two arms (levers). Upon pulling an arm, one receives a reward of either 0\$ or 1\$. This translates to an MDP with one state and two actions, i.e., $A = \{\leftarrow, \rightarrow\}$ where "$\leftarrow$" means "pull the left arm" and "$\rightarrow$" means "pull the right arm". The reward set is $R = \{0, 1\}$. The MDP dynamics are determined by the two probabilities $p(0|\leftarrow)$ and $p(0|\rightarrow)$ of losing, i.e., "winning" 0\$, when pulling the left or right arm. Each policy is determined by $\pi(\leftarrow)$ which denotes the probability of choosing the left arm. The learning problem is to identify the arm that yields the highest value $v^*$, given by $v^* = \max\{p(1|\leftarrow), p(1|\rightarrow)\}$.

We encode the actions via $|\leftarrow\rangle = |0\rangle_A$ and $|\rightarrow\rangle = |1\rangle_A$ which can be done by using single qubit. We use another qubit to encode the rewards as $|0\$\rangle = |0\rangle_R$ and $|1\$\rangle = |1\rangle_R$. In this encoding, we can represent the step operator $\mathbf{S}$ as a quantum circuit shown in Figure 2. The angles of the $Y$-axis rotations $R_y$ are given by $\theta^\pi = 2\arccos\left(\sqrt{\pi(\leftarrow)}\right)$, $\theta^\leftarrow = 2\arccos\left(\sqrt{p(0|\leftarrow)}\right)$ and $\theta^\rightarrow = 2\arccos\left(\sqrt{p(0|\rightarrow)}\right)$. A direct calculation shows that this $\mathbf{S}$ indeed prepares a qsample of one agent-environment interaction.

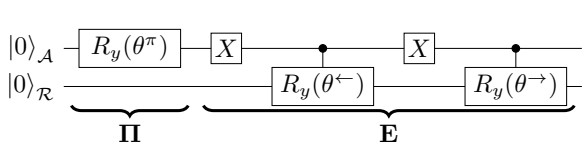

Figure 2: The step operator $\mathbf{S}$ for the two-armed bandit MDP in the form of a quantum circuit.

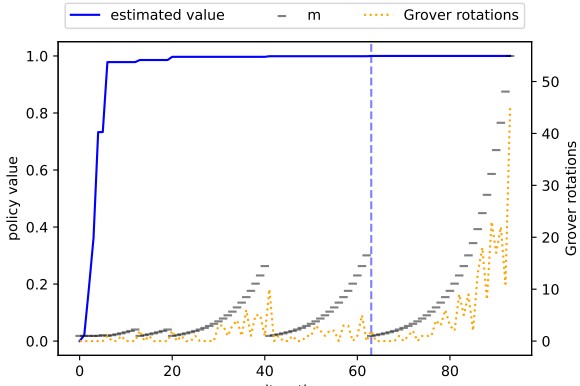

Figure 3: Distribution of the $\tilde{v}_\pi^2$ output of QPE for the concrete instance of the two-armed bandit MDP.

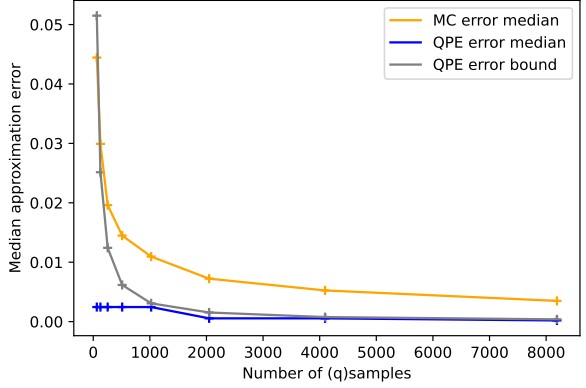

Figure 4: Median approximation errors of 1000 runs of both QPE and classical MC for one round of two-armed bandit.

Figure 5: The estimated policy value $v_k$ and the Grover rotations applied during one successful run of quantum policy iteration.

Now consider the two-armed bandit for horizon $H = 2$, i.e., the agent gets to play two rounds. The MDP operator of this decision process is given by two step operators according to Figure 2 that act on two separate $\mathcal{A} \otimes \mathcal{R}$ subsystems represented by four qubits. For simplicity, we set the discount factor $\gamma = 1$. The return operator $\mathbf{G}$ adds the two reward qubits and stores the result using another two-qubit subsystem that represents the return space $\mathcal{G}$. It can be implemented using two CNOT gates and one Toffoli gate (which is a CNOT gate with two control qubits) to realize the logical expressions that determine which qubit of the return register must be set to $|1\rangle$ or $|0\rangle$ depending on the rewards. The gate $\mathbf{\Phi}$ can be realized using three doubly-controlled $R_y$ gates that rotate another ancillary qubit depending on the two reward qubits.

## 5.1 Simulations of Quantum Policy Evaluation

Putting all of the above together, we receive a gate-decomposition of $\mathbf{A}_{\mathrm{QPE}}^\phi$. To simulate QPE, we implemented the operator in IBM's `qiskit` (Aleksandrowicz et al., 2019) framework for the programming language `Python`. We chose a concrete bandit with dynamics $p(0|\leftarrow) = 0.55$ and $p(0|\rightarrow) = 0.65$ and considered the policy $\pi$ given by $\pi(\leftarrow) = 0.50$ which has the value $v_\pi^2 = 0.80$. All of these values were chosen arbitrarily. We used simulated QPE to obtain an estimate of this value with absolute error of at most $\epsilon = 0.025$. As maximum error probability, we chose $\delta = 0.05$. According to Equations (31) and (36), we have to use

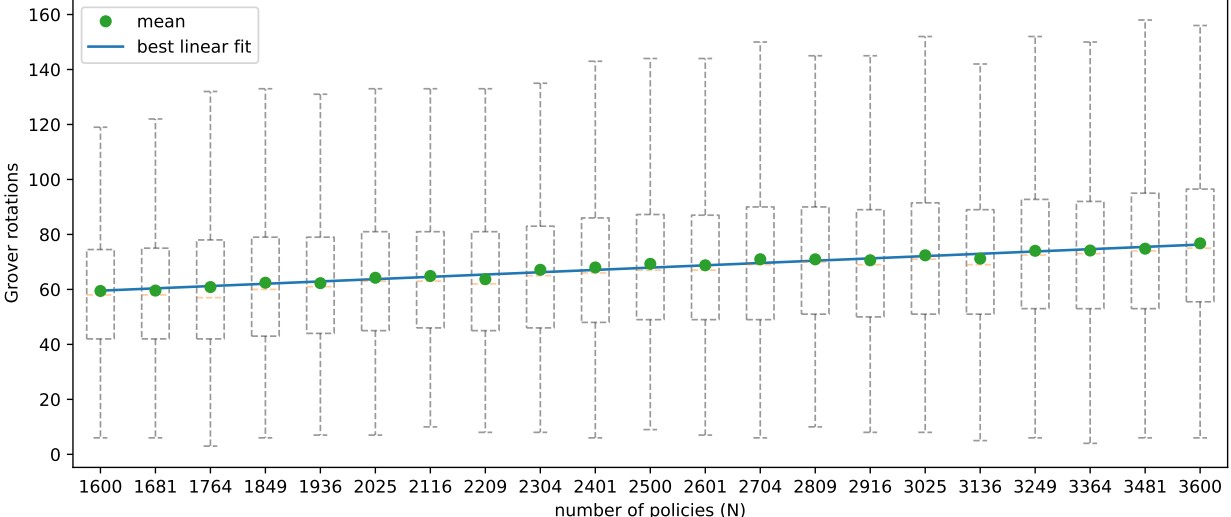

Figure 6: The total numbers of Grover rotations until quantum policy iteration finds an $\epsilon$-optimal policy for the bandit problem.

$t = 7 + 4 = 11$ qubits in the first register of the phase estimation circuit to achieve the $\epsilon$-approximation with probability at least $1 - \delta$. The distribution of the QPE outputs for these parameters is shown in Figure 3. Qiskit offers the possibility to calculate measurement probabilities analytically, which is what we used to generate the plot. The probability distribution of the QPE output $\tilde{v}_\pi^2$ is concentrated at the true value at $v_\pi^2 = 0.80$ and the mass that lies outside the $\epsilon$ region is less than $0.025$ which is even better than our guaranteed error probability bound of $\delta = 0.05$.

Next, we wanted to empirically confirm the quantum advantage of QPE over classical MC. To this end, we used both methods to evaluate a fixed policy for a fixed bandit using an increasing number of (q)samples. To reduce computational complexity, we chose horizon $H = 1$. According to Equation (36), the approximation error $|v_\pi^1 - \tilde{v}_\pi^1|$ of QPE is in $\mathcal{O}(1/2^n)$, where $n$ is from the definition of $t$ in Equation (31). For our experiment, we set $t = n$. From Figure 1, we see that QPE then uses $2^{n+1} - 1$ qsamples via applications of $\mathbf{A}_{\text{QPE}}$ and $\mathbf{A}_{\text{QPE}}^\dagger$. We chose a range of integer values for $n$ and for each $n$ executed QPE 1000 times and calculated the median approximation error. Recall that the failure probability for QPE with $t = n$ is $8/\pi^2$ so taking the median delivers a run in which the algorithm indeed returned an $\epsilon$-approximation of the true value with high probability. The results of this experiment are shown in Figure 4. We also included the error bound $\epsilon$ according to Equation (36) and see that the approximation errors satisfy the theoretical guarantee. For each $n$, we also collected $2^{n+1} - 1$ classical samples, averaged them, repeated this 1000 times and calculated the median. The approximation error of this classical MC approach is always higher than the one of QPE. This empirically confirms the quantum advantage of QPE over MC in terms of (q)sample complexity.

## 5.2 Simulations of Quantum Policy Iteration

We now turn to policy iteration. As a toy learning problem, we used a two-armed bandit where the agent always loses when it chooses the left arm and always wins when it chooses the right arm, i.e., $p(0|\leftarrow) = 1$ and $p(1|\rightarrow) = 0$. In DP, one typically only considers deterministic policies as, under mild assumptions, every finite MDP has an optimal deterministic policy (Sutton & Barto, 2018). For the two-armed bandit, there are only two such policies, which results in an uninteresting problem. To make the search for an optimal decision rule more challenging, we instead used a set of $N \in \mathbb{N}$ stochastic policies given by

$$\mathsf{P}_N = \left\{ \pi^n : \pi^n(\leftarrow) = \frac{n-1}{N-1}; n = 1, ..., N \right\}. \tag{42}$$

We let the agent start with the worst policy $\pi^1$ which chooses the left arm with unit probability and want to find the optimal decision rule $\pi^* = \pi^N$.

Figure 5 documents one successful run of quantum policy iteration for the toy learning problem with $N = 1000$ policies and parameters $\epsilon = 0.0125$, $\delta = 0.07$, $C = 30$ and $\lambda = 8/7$. We set $H = 1$ so the agent plays one round and the value of the optimal policy is $v^* = 1$. In the beginning, the policy value rapidly increases even when no or only few Grover rotations are applied. This is because the agent starts with the worst possible policy, and better policies are easily found by chance. The steep increase stops after a few iterations, when the policy is close to optimal. It takes increasingly more rotations, i.e., amplifications of the then small success probability of finding a better policy, and some minor improvements until the procedure makes the final jump marked by the dotted vertical line. After that, the values stagnate at the maximum of 1 as from then on, no further improvement is possible. During the last iterations, the algorithm tries more and more Grover rotations as $m$ (that determines their maximum number) grows exponentially.

Finally, we investigated our conjecture that the run-time of quantum policy iteration is proportional to $\sqrt{|\mathsf{P}|}$. To empirically test this, we ran quantum policy iteration for the same bandit and using the same parameters as in the previous experiment. We quadratically increased the size $N$ of the policy set $\mathsf{P}_N$ from $N = 40^2 = 1600$ to $N = 60^2 = 3600$. For each $N$, we ran quantum policy iteration 1000 times and calculated statistics of the resulting distribution of the number of Grover rotations. As the number of rotations depends on the patience $C$, we omitted the ones that happened in the last $C$ iterations. We only considered "successful" runs where quantum policy iteration returned an $\epsilon$-optimal policy. Due to the low QPE failure probability, more than 99% of all runs were successful for any $N$. The results of the complexity experiment are shown in Figure 6. We see that the average number of rotations indeed seems to grow linear with $\sqrt{N}$, i.e., quadratically with the number of policies $N$. The line through the means was found by linear regression and fits the data well; the mean squared error is 1.50. Note that for each $N$, the empirical distribution of the runtimes (we hid the outliers for the sake of clarity), appears symmetric and has a large interquartile range. This is expected because sometimes, the algorithm samples a near optimal policy early by chance while in more unlucky runs, the optimum is found through many incremental improvements.

## 6 Conclusion

In this work, we described a novel quantum realization of a classical MDP. Based on this construction, we showed in detail how to combine amplitude estimation and Grover search to obtain a quantum algorithm that realizes the classical scheme of policy evaluation and improvement. This is a concrete proof-of-concept for how to solve reinforcement learning problems exclusively on quantum computers. We showed that by leveraging the superposition principle, we can reduce the sample complexity of classical MC policy evaluation. This indicates that quantum advantage in RL might be achieved by lowering the number of agent-environment interactions it takes for an agent to find the optimal policy.

In their current form and given the likely limitations of near-term quantum hardware, our methods are not applicable to real problems: The qubit and gate complexity of QPE scales linear with the horizon and quantum policy iteration requires enough qubits to enumerate all possible policies of a finite MDP. Moreover, the quantum advantage of QPE over classical MC-based policy evaluation holds only in terms of sample complexity and only if efficient, error-free implementations of policy and environment operators are readily available. These operators prepare qsamples of the agent's policy and the MDP dynamics, which are then used for amplitude estimation in QPE. Herbert (2021) showed that the quadratic quantum advantage of amplitude estimation over classical MC can be eradicated if the preparation of the distribution as a quantum state, that is in our case performed by the policy and environment operators, is not efficient enough. However, more recently, the same author proposed another method for quantum MC that also yields a quadratic speedup, but may not suffer from efficiency problems with state preparation (Herbert, 2022). The investigation, if and how this method can be integrated into our framework as a sub-routine of QPE is left for future work. Another possible direction for future work would be to try to improve the policy improvement step. Note that QPE can be integrated with other quantum optimization methods than Grover-searches. It might also be possible to combine QPE with quantum versions of dynamic programming which would be more efficient than directly searching for an optimal policy.

## Acknowledgment

The project this report is based on was supported with funds from the German Federal Ministry of Education and Research in the funding program *Quantum Technologies - From Basic Research to Market* under project number 13N15644. The sole responsibility for the report's contents lies with the authors.

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

## A    Proof of the error bound of QPE

In this appendix, we prove Equations (35) and (36). The proof uses the following technical result:

**Lemma 1** ((Brassard et al., 2002), Lemma 7). *Let $\mu = \sin^2(\alpha)$ and $\tilde{\mu} = \sin^2(\tilde{\alpha})$ with $0 \leq \alpha, \tilde{\alpha} \leq 2\pi$. Then it holds that*

$$|\tilde{\alpha} - \alpha| \leq a \Rightarrow |\tilde{\mu} - \mu| \leq 2a\sqrt{\mu(1-\mu)} + a^2. \tag{43}$$

To see that Equation (35) is true, recall from Equation (32) that the random variable $X$ that corresponds to the final measurements of phase estimation in QPE satisfies

$$P\Big(\big\{|X/2^t - \theta| \leq 1/2^{n+1}\big\} \cup \big\{|X/2^t + \theta| \leq 1/2^{n+1}\big\}\Big) \geq 1 - \delta. \tag{44}$$

We set $\tilde{\mu} := \sin^2(\pi X/2^t) = \phi\big(\tilde{v}_\pi^H(s)\big)$ and $\mu := \sin^2(\pi\theta) = \phi\big(v_\pi^H(s)\big)$ and obtain

$$P\left(|\tilde{\mu} - \mu| \leq \frac{\pi}{2^{n+1}} + \frac{\pi^2}{2^{2n+2}}\right) \geq P\left(|\tilde{\mu} - \mu| \leq \pi\frac{\sqrt{\mu(1-\mu)}}{2^n} + \frac{\pi^2}{2^{2n+2}}\right) \geq 1 - \delta, \tag{45}$$

where the first inequality holds as $\sqrt{\mu(1-\mu)} \leq 1/2$ and the second inequality is due to Lemma 1. Now the form of $\epsilon$ as stated in Equation (36) follows from

$$\big|\tilde{v}_\pi^H(s) - v_\pi^H(s)\big| = \big|\phi^{-1}(\tilde{\mu}) - \phi^{-1}(\mu)\big| = (\bar{g} - \underline{g})|\tilde{\mu} - \mu|, \tag{46}$$

where in the last step, we used the definition of $\phi$.

