# OpenReview forum: "Quantum Policy Iteration via Amplitude Estimation and Grover Search – Towards Quantum Advantage for Reinforcement Learning"
_TMLR — Accepted by TMLR_

### Review · Reviewer_4dsk · 2022-12-15

**Summary Of Contributions:**

The authors show a method for constructing quantum algorithms for policy evaluation and policy improvement.
They combine some well-known quantum algorithms to achieve their aim, with the main contribution being the construction of unitaries that can achieve policy evaluation.
They then numerically evaluate their proposed algorithm via simulated circuits, demonstrating good evidence for their claims that the classical approximation error is larger than the quantum approximation error for a given number of samples and that the number of Grover rotations required scales as $\sqrt{|P|}$.

**Audience:**

Yes

**Claims And Evidence:**

No

**Requested Changes:**

- The authors acknowledge that error correction will be required to run this algorithm, but there is no discussion of the overheads of error correction, and at what accuracy level the quadratic speedup from Quantum Monte Carlo could overcome these. A full T-gate complexity costing would demonstrate this, but is not expected for a single review.

- There is no discussion of the resources required to build the $\Pi$, E, S, M operators, i.e. loading the probability distribution to sample from into the quantum state and the errors that would be produced from this, see [Herbert 2021](doi.org/10.1103/PhysRevE.103.063302) for a discussion of the scaling of Monte Carlo in the presence of state preparation errors. The authors should demonstrate that this algorithm is not subject to such errors if a quantum speedup is available.

- In section 3.3 they claim that the probability of a non-policy-improving error can be limited by choosing small values for $\delta$, but they fail to discuss that this is related to the number of qubits required for phase estimation, which in turn scales exponentially with the number of calls to the $Q_{\textrm{QPE}}$ oracle. I am aware that the logarithm of $1 / \delta$ is taken for $t$, so an approximately linear dependence on the number of calls required, but reducing $\delta$ does not come for free.

- Definition of amplitude estimation in Section 3.2 is not novel and could be moved to an appendix (I do agree it would be useful for the audience of this journal).

- The acronym QPE is already heavily in use referring to Quantum Phase Estimation, consider changing to avoid a clash.

- The authors of qiskit request that it is cited via a Bibtex file found on the Github repo.

- The claim of quantum speedup is not accurate without addressing these concerns, consider weakening the claim if a full resource estimation is not possible.


**Strengths And Weaknesses:**

### Strengths:

- Construction of unitaries to achieve the task of Quantum Policy Evaluation, and use of standard quantum algorithms (Monte Carlo, Phase Estimation and Grover's search) for QPE, and its extension to Quantum Policy Improvement.

- Numerical evidence for the sampling and accuracy scaling claims.

- Sharing of code used for numerics.

### Weaknesses:

- The authors claim "provable [Quantum] speedups" in the abstract, whereas they have not addressed all of the resource requirements for this algorithm, only sampling complexity. A more detailed discussion is in the below section.

- Some of the quantum algorithms used are unmodified from the original papers, whereas there have since been improvements that reduce resources, e.g. see Table 1 of [Babbush et. al 2018](doi.org/10.1103/PhysRevX.8.041015) for phase estimation improvements. I would not expect changes to address this.

---

> ### Author Response · Authors · 2023-01-02
> **Authors' Response**
>
> Thank you very much for your review. Here are our answers to your points:
>
> - `There is no discussion of the resources required to build the Π, E, S, M operators, i.e. loading the probability distribution to sample from into the quantum state and the errors that would be produced from this [...].`
>
> - `The claim of quantum speedup is not accurate without addressing these concerns [...]`
>
> You are right, our work does not include such a discussion. The algorithms and results assume that there is access to all these operators that are needed to realize the MDP and that they "work" error-free. We added the following statement to the conclusion to emphasize this:
>
> `"[W]e assumed an ideal setup in which policy and environment operators are readily available and error-free and the improvement in sample complexity of QPE over classical MC only holds if this is the case."`
>
> Moreover, we followed your suggestion and made the claim in the abstract more specific. We changed the last sentence to:
>
> `Our work is a detailed and formal proof of concept for how
> quantum algorithms can be used to solve RL problems and shows that they can indeed yield
> provable improvements over classical MC based methods in terms of sample complexity.`
>
> - `In section 3.3 they claim that the probability of a non-policy-improving error can be limited by choosing small values for `$\delta$`, but they fail to discuss that this is related to the number of qubits required for phase estimation, which in turn scales exponentially with the number of calls to the `$Q_{\text{QPE}}$` oracle.`
>
> We added the statement that decreasing delta `"comes at the cost of increasing the qubit complexity and run-time of QPE."`
>
> - `The authors of qiskit request that it is cited via a Bibtex file found on the Github repo.`
>
> Thank you for pointing that out, we adapted the citation accordingly.

---

> > ### Comment · Reviewer_4dsk · 2023-01-09
> > **Comment**
> >
> > Thank you for your comments and edits. I feel that the assumption that these operators work error free is a very strong assumption, and your scheme is vulnerable to "de-quantusation". I may have mis-represented the conclusions of Herbert2021, as it is not errors in the operators that concerns me (as this would be ran on a fault tolerant device), but the circuit depth required to load any useful probability distribution would erase any quantum speedups. Nothing is shown to persuade me that the input data can be loaded efficiently, and this may be misleading without explicit discussion of the pitfalls of classical data loading.
> > Reduction of the claim goes some way to alieviating this concern.
> >
> > Thank you for clarifying the discussion around $\delta$.

---

> > > ### Author Response · Authors · 2023-01-10
> > > **Changes**
> > >
> > > Thank you very much for the clarification. We added the following discussion to the conclusion and hope that this clarifies the nature of the quantum advantage of QPE and the conditions that have to be satisfied:
> > >
> > > `[T]he quantum advantage of QPE over classical MC-based policy evaluation holds only in terms
> > > of sample complexity and only if efficient, error-free implementations of policy and environment operators
> > > are readily available. These operators prepare qsamples of the agent’s policy and the MDP dynamics,
> > > which are then used for amplitude estimation in QPE. Herbert (2021) showed that the quadratic quantum
> > > advantage of amplitude estimation over classical MC can be eradicated if the preparation of the distribution
> > > as a quantum state, that is in our case performed by the policy and environment operators, is not efficient
> > > enough. However, more recently, the same author proposed another quantum method for quantum MC that
> > > also yields a quadratic speedup, but may not suffer from efficiency problems with state preparation (Herbert,
> > > 2022). The investigation, if and how this method can be integrated into our framework as a sub-routine of
> > > QPE is left for future work.`
> > >
> > > We also made the conditions clearer in the abstract:
> > >
> > > `Our work is a detailed and formal proof of concept for how
> > > quantum algorithms can be used to solve RL problems and shows that, given access to error-
> > > free, efficient quantum realizations of the agent and environment, quantum methods can
> > > yield provable improvements over classical MC based-methods in terms of sample complexity.`
> > >
> > > We hope that these changes address your concern sufficiently.

---

### Review · Reviewer_Hrjz · 2022-12-19

**Summary Of Contributions:**

The authors propose a quantum based method for finding the optimal policy (QPI). The algorithm is based on the concept of policy iteration, which involves alternating between two steps: policy evaluation and policy improvement.
The QPI algorithm works by representing the states, actions, and rewards as quantum states, and using unitary operators to model the interactions between the agent, the environment, and the rewards. These operators can be implemented on a quantum computer, allowing for the efficient computation.

In the policy evaluation step, the value function is calculated for all states under the current policy. This is done by using the policy operator and the environment operator to simulate the interactions between the agent, the environment, and the rewards, and summing up the expected reward for each state. This is effectively a quantum version of simple Monte-Carlo (MC).

In the policy improvement step, the policy is updated in order to improve the expected reward. This is done by first mapping all deterministic policies into the earlier policy operator space and then using quantum search on the resulting function.

Finally they show simulation results of their algorithm on a 2-armed bandit.

**Audience:**

No

**Broader Impact Concerns:**

I do not see any significant ethical implications from this work.

**Claims And Evidence:**

No

**Requested Changes:**

While the authors have explained their differences with methods like Wang et al (2021) and Duchko et al (2016). Specifically there are earlier works on directly using Grover search and Amplitude estimation for stochastic planning (Naguleswaran and White, 2005). Rosenwald et al (2004) also have used Grover search for control purposes on POMDP. More recently, Cherrat et al have also proposed a quantum version of policy iteration.

As mentioned earlier a more MDP-esque experimental setup would be important to claim any practical improvements over classical RL methods. The current set of experiments are uninformative in this regard.

I also would generally like a method which combines quantum ideas with some classical RL ideas, or at the very least not consider it as a blind monte carlo estimator. For example, while the authors show that QPE has a better than classical sample complexity, the classical bounds applied dont hold for RL setting (for example Jiang and Li, 2015)

-----------------------------------------------
Quantum search in stochastic planning, S. Naguleswaran and L. White, 2005

Applications of Quantum Algorithms to Partially Observable Markov Decision Processes, R. Rosenwald, D. Meyer, and H. Schmitt, 2004

Quantum Reinforcement Learning via Policy Iteration E. Cherrat, I. Kerenides, A. Prakash, 2022

Doubly Robust Off-policy Value Evaluation for Reinforcement Learning, N. Jiang and L. Li, 2015


**Strengths And Weaknesses:**

Strengths:

The paper is easy to read and follow. They introduce and describe the quantum versions of the required classical objects well, and someone non familiar with quantum computing should be able to broadly follow the paper.

The premise of demonstrating quantum advantage directly on policy search interesting.

The authors show that quantum methods can improve over MC based methods of policy evaluation

Weaknesses:

The proposed method has not got any insight into the standard RL setup. It ignores the entire dynamic programming/Bellman equation based framework of RL, and instead uses simple MC estimation.  Furthermore the bounds used by the authors to show quantum advantage do not hold for MDPs when one uses the dynamic programming/bellman methods.

Experiments are on a bandit which is not really an MDP. While I agree that simulating large circuits is difficult; the current experimental setup  throws away any advantage of classical RL methods, as for a bandit they become some version of simple MC. I would have liked to see atleast a horizon of length 3, so that the classical techniques for MDP can actually be useful.

I am not sure how much I can agree with the quantum MDP construction. Specifically since the transition and reward probabilities are directly encoded directly in the phase of the quantum state, in principal one can use amplitude estimation directly to compute the probabilities. If the model of the MDP is known, then the problem is significantly easier in the classical setting as well.

---

> ### Author Response · Authors · 2023-01-02
> **Authors' Response**
>
> Thank you very much for your review. We are happy that you find our work interesting and easy to understand. To address your concerns, we allowed us to partition them into groups of points that we found to be related:
>
> - `The proposed method [...] ignores the entire dynamic programming/Bellman equation based framework of RL [...].`
>
> - `[T]he bounds used by the authors to show quantum advantage do not hold for MDPs when one uses the dynamic programming/bellman methods.`
>
> - `If the model of the MDP is known, then the problem is significantly easier in the classical setting as well.`
>
> - `As mentioned earlier a more MDP-esque experimental setup would be important to claim any practical improvements over classical RL methods. The current set of experiments are uninformative in this regard.`
>
> - `I also would generally like a method which combines quantum ideas with some classical RL ideas, or at the very least not consider it as a blind monte carlo estimator.`
>
> - `For example, while the authors show that QPE has a better than classical sample complexity, the classical bounds applied dont hold for RL setting [...]`.
>
> The aim of our work was not to develop a quantum reinforcement learning framework that is in any way superior to the standard approaches based on dynamic programming. Rather than that, we wanted to provide a detailed proof-of-concept that one can solve a classical MDP using exclusively quantum methods given that one has "quantum access" to samples of the MDP's dynamics (that is in our case provided by the policy and environment operator). Therefore, we also did not include theoretical or "practical" comparisons of the quantum methods versus dynamic programming or any other approach. We acknowledge that our methods do not exploit results from dynamic programming and do not have an advantage over them. The quantum advantage we showed for QPE holds only compared to classical MC based policy evaluation.
>
> - `[T]here are earlier works on directly using Grover search and Amplitude estimation for stochastic planning (Naguleswaran and White, 2005).`
> - `[C]herrat et al have also proposed a quantum version of policy iteration.`
>
> Thank you very much for pointing out this reference. We added a brief discussion of these two  papers to Section 2 ("Related Work").

---

> > ### Comment · Reviewer_Hrjz · 2023-01-12
> > **Comment**
> >
> > Thank you for the response and edits to the paper.
> > I think I would still like to see some results in an RL setting compared to bandit settings. Direct MC evaluation of an MDP is essentially a bandit problem and there already exist quantum algorithms with strong guarantees in this area (see Casale et al). The specific quantum MDP construction is a minor extension of the bandit operator described in earlier works. While a theoretical investigation might be outside the scope of current work, an evaluation on MDP setup is in my opinion important.
> >
> > Casale, B.; Di Molfetta, G.; Kadri, H.; and Ralaivola, L. 2020. Quantum Bandits

---

> > > ### Author Response · Authors · 2023-02-07
> > > **Authors' Response**
> > >
> > > Regarding the "Bandit vs. MDP" question raised by you and the action reviewers, we hope that the following thoughts and remarks address all concerns:
> > >
> > > We fully acknowledge that the two-armed bandit MDP with its single state and two actions is a very simple problem that has little practical relevance. We, too, would have liked to present simulations for larger common benchmark problems, such as for example a stochastic maze environment. Unfortunately, we are confronted with the problem that quantum circuits are hard to simulate on classical machines, especially if they operate on a lot of qubits, which is the case for our methods. Just an example: For a 10x10 maze with 4 actions and horizon H, there are 400^H trajectories which have to be encoded in a single statevector in order to accurately simulate QPE. Processing such enormous arrays is infeasible already for small H. However, we find that the implementation for the two-armed bandit still adds good insight into how our methods work in principle, as all operators of the quantum MDP have intuitive implementations. Finally, we would like to emphasize that our methods are designed to work on any MDP and are not specific to bandits.
> > >
> > > Regarding your concern that `[d]irect MC evaluation of an MDP is essentially a bandit problem`, we agree that a direct (Grover) search to find the optimal policy is arguably not be the best approach to solve an MDP. The aim of our work was to provide a detailed algorithm to solve a general MDP on a gate-based quantum computer that yields a certain quantum advantage. Our QPE method has this certain quantum advantage over classical Monte Carlo. We chose to combine it with the Grover-based optimization over all policies to provide a complete method to solve the MDP. QPE can also be used as sub-routine for other quantum RL methods. Exploring it together with other means for policy improvement to arrive at a more efficient routine could be an interesting direction for future work.

---

### Review · Reviewer_MaGx · 2022-12-20

**Summary Of Contributions:**

This paper presents quantum analogues of classical policy evaluation and policy improvement algorithms for quantum mechanical realizations of finite Markov decision processes, leveraging existing quantum techniques such as amplitude estimation and Grover search. As a proof-of-concept, they present empirical evaluations of their method on a simulated quantum process of a two-armed bandit problem.

**Audience:**

Yes

**Claims And Evidence:**

No

**Requested Changes:**

I think for a paper like this to be accepted in TMLR it would require significant rewriting, and would need to elaborate the quantum concepts much more slowly, and connecting them to terms the RL community is familiar with.

I'd encourage the authors to consider resubmitting a (much more) expanded version of this work that is more accessible to the RL/ML community.

**Strengths And Weaknesses:**



At a very high-level, this work seems interesting and quite forward-looking (as the authors themselves point out, "Existing quantum hardware is not yet ready to run QPE, let alone quantum policy iteration").

However, given that it was submitted to TMLR, my impression is that the main audience reading this will be the RL/ML community. With this in mind, I feel this paper is too detached from these communities. The authors seem to rely on lots of quantum background which most readers of this publication will lack. To contextualize my comments below: I have been working in RL for over 15 years, so I feel my understanding (or lack thereof) is likely a reasonable reflection of that of the larger community.

I think for a paper like this to be accepted in TMLR it would require significant rewriting, and would need to elaborate the quantum concepts much more slowly, and connecting them to terms the RL community is familiar with.

Starting from section 3.1 things are already difficult to follow. The maps presented in equations 1 - 5 seem to simply be rewriting a standard MDP definition, but using quantum notation. Is this so? What is the difference? For instance, is equation 5 just a rewriting of the standard MDP transitions or is there a difference? For someone not familiar with quantum terminology, this needs more contextualization.

Section 3.2 was basically totally incomprehensible to me.

One last comment is that it wasn't totally clear to me how much of this paper was novel, and how much was reusing prior work. In particular, it seems to be leveraging results from Brassard et al. (2002) a fair bit. It would be good if the authors could also provide more clarity on the parts of the paper that are novel, and the parts that are reusing prior work.

---

> ### Author Response · Authors · 2023-01-02
> **Authors' Response**
>
> Thank you very much for your review.
>
> **Regarding your main concern about the understandability of our work:** A key element of our paper is to combine reinforcement learning with quantum computing. We fully acknowledge that to understand our work in full detail, one needs to have strong familiarity with concepts and notions used in both fields. We chose to omit more thorough introductions to such aspects to keep our work as concise as possible for readers that have a background in both reinforcement learning and quantum computing. The paper is the distilled version of a thesis and given your concerns, we are considering to make the thesis publicly available and to include it as a reference.
>
> **Regarding your questions about Section 3.1:** In Section 3.1, we re-define the components of a classical MDP (states, actions, rewards,  transitions) in the language of quantum mechanics. The way we designed our definitions is such that all concepts "remain the same" and that nothing is added/left out. The key difference between the quantum MDP and the classical one is that the quantum version uses entanglement to model the probabilistic nature of the classical MDP.
>
> **Regarding your question about the novelty of our work:** The key contributions of our work are to define a quantum analogue of the classical MDP, to provide a proof of concept that the quantum version can be solved using only quantum methods and that these methods are more sample-efficient than a comparable classical method. For the last two contributions, we substantially relied on existing quantum algorithms such as phase estimation and Grover search.

---

> > ### Comment · Reviewer_MaGx · 2023-01-09
> > **Background**
> >
> > Thank you for your response. Adding a pointer to the thesis with a more thorough background I think would be a great idea, as it avoids making the current submission too long, while still providing the necessary background for those who lack it.
> >
> > Thank you for clarifying the points about the MDP redefinition and the novelty. I think it would be good to clarify these points in the submission.

---

> > > ### Author Response · Authors · 2023-01-10
> > > **Changes**
> > >
> > > For further clarification, we added the following paragraph to the beginning of Section 3.1:
> > >
> > > `In this section, we develop a quantum version of the classical MDP. The main difference between our construction and the classical one is that we use entanglement to model the stochasticity of the agent-environment interaction. Everything else is designed to be completely analogous to the classical case.`
> > >
> > > We also decided to make the thesis publickly available if ther paper is accepted. Doing so now would reveal our identity.
> > >
> > > We hope that this fully addresses your concerns.

---

### Author Response · Authors · 2023-02-16
**Revision**

Dear Action Editors, Dear Reviewers,

we have just uploaded an updated version of our paper. As suggested, we rewrote parts of the introduction and conclusion to make our contributions clearer. Moreover, we added a "Background" section in which we discuss the most important concepts from quantum computing that we use in our work.
We would again like to thank all of you for your suggestions and we believe that the changes during the review process made our message clearer and our work more accessible to a broader audience in the TMLR community.

---

> ### Comment · Reviewer_4dsk · 2023-03-02
> **Comment on background**
>
> Thanks for adding the clear background on quanutm algorithms, it will help for the audience of this paper, and stays relevant to what you present later.
>
> My only suggestion is that as a description of quantum algorithms it is missing a large element behind the power of quantum algorithms, entanglement. It is possible to understand the paper as is, but in both the quantum phase estimation step, and the ansatz circuit, you use entangling gates.
>
> I believe you can improve the background again with a (very brief) discussion of entangling gates; i.e. in the QPE discussion you describe the controlling gate as applying if and only if the control qubit is $| 1 \rangle$, this is possibly not clear for readers new to the quantum algorithms concepts on the action of the gate if the control qubit is in a superposition state e.g. $|+\rangle$.

---

> > ### Author Response · Authors · 2023-03-03
> > **Thank you for your suggestion**
> >
> > We are happy that you find the background section useful and thank you for your suggestion. It should be possible to add a paragraph on entanglement that fits in with the rest of the background. We will make this addition and upload a revised version shortly.

---

### Decision · Action_Editors · 2023-03-05

**Recommendation:** Accept with minor revision

**Comment:**

Two experts in quantum RL were found to review this paper. In addition a senior RL researcher with no background in quantum was also brought in. During the first round issues with the utility of the bandit experiments were raised, but most critically the senior RL reviewer and the AC found the submission extremely difficult to understand.

The authors added a significant background section, and further contextualisation of the bandit experiment. The AC and two reviewers voting yes for acceptance.

There are some outstanding issues:
1)  add a short reference to section 3 in the intro, as it starts off talking about "unitary operators" and "exchange states"
2) add entanglement into their background
3) make sure the justifications about the bandit experiments is fully integrated into the paper.

**Audience:**

The paper contains a nice introduction to quantum concepts providing a pathway for an RL researcher to understand the contents. Good fit for the TMLR community.

**Claims And Evidence:**

This investigates quantum policy iteration introducing new algorithms, numerical simulations on a bandit task, and perhaps most importantly serves as a nice introduction to quantum RL. Reviewers agreed this is a solid piece of work.

---

> ### Author Response · Authors · 2023-03-06
> **Thank you**
>
> We are very happy to receive these news! Thank you and all reviewers for your constructive feedback and suggestions that improved our paper and made it more accessible.
>
> We will make all requested changes and will upload the camera ready version of the paper shortly.

---

> ### Author Response · Authors · 2023-03-14
> **Camera Ready**
>
> We have just uploaded the camera ready version of the paper including all requested changes.